# Circulating Extracellular miRNA Analysis in Patients with Stable CAD and Acute Coronary Syndromes

**DOI:** 10.3390/biom11070962

**Published:** 2021-06-29

**Authors:** Andrey V. Zhelankin, Daria A. Stonogina, Sergey V. Vasiliev, Konstantin A. Babalyan, Elena I. Sharova, Yurii V. Doludin, Dmitry Y. Shchekochikhin, Eduard V. Generozov, Anna S. Akselrod

**Affiliations:** 1Department of Molecular Biology and Genetics, Federal Research and Clinical Center of Physical-Chemical Medicine of Federal Medical Biological Agency, 119435 Moscow, Russia; babalyan@rcpcm.org (K.A.B.); sharova.ei@rcpcm.org (E.I.S.); generozov@rcpcm.org (E.V.G.); 2Department of Cardiology, Functional and Ultrasound Diagnostics, Faculty of Medicine N.V. Sklifosovsky, I.M. Sechenov First Moscow State Medical University of the Ministry of Health of the Russian Federation (Sechenov University), 119146 Moscow, Russia; stonogina.d@gmail.com (D.A.S.); lod.kain@gmail.com (S.V.V.); agishm@list.ru (D.Y.S.); 7402898@mail.ru (A.S.A.); 3FSI National Research Center for Preventive Medicine of the Ministry of Health of the Russian Federation, 101990 Moscow, Russia; ydoludin@gnicpm.ru

**Keywords:** acute coronary syndrome, circulating microRNA, coronary heart disease, non-invasive biomarker

## Abstract

Extracellular circulating microRNAs (miRNAs) are currently a focus of interest as non-invasive biomarkers of cardiovascular pathologies, including coronary artery disease (CAD) and acute coronary syndromes (ACS): myocardial infarction with and without ST-segment elevation (STEMI and NSTEMI) and unstable angina (UA). However, the current data for some miRNAs are controversial and inconsistent, probably due to pre-analytical and methodological variances in different studies. In this work, we fulfilled the basic pre-analytical requirements provided for circulating miRNA studies for application to stable CAD and ACS research. We used quantitative PCR to determine the relative plasma levels of eight circulating miRNAs that are potentially associated with atherosclerosis. In a cohort of 136 adult clinic CAD patients and outpatient controls, we found that the plasma levels of miR-21-5p and miR-146a-5p were significantly elevated in ACS patients, and the level of miR-17-5p was decreased in ACS and stable CAD patients compared to both healthy controls and hypertensive patients without CAD. Within the ACS patient group, no differences were found in the plasma levels of these miRNAs between patients with positive and negative troponin, nor were any differences found between STEMI and NSTEMI. Our results indicate that increased plasma levels of miR-146a-5p and miR-21-5p can be considered general ACS circulating biomarkers and that lowered miR-17-5p can be considered a general biomarker of CAD.

## 1. Introduction

Coronary artery disease (CAD) is the leading cause of cardiovascular mortality worldwide [1,2,3]. Acute coronary syndromes (ACS) constitute a subcategory of CAD and occur as a result of atherosclerotic plaque disruption and thrombotic occlusion of the artery. ACS subtypes include unstable angina (UA), acute myocardial infarction (MI) with ST-segment elevation (STEMI), or MI without ST elevation (NSTEMI)—all of which are conditions representing myocardial ischemia or injury. Patients with stable CAD are those whose symptoms are adequately controlled through medical therapy. However, some of them develop unpredictable complications and require percutaneous or surgical intervention [4]. 

To date, several biomarkers have been discovered and investigated that make it possible to diagnose myocardial damage and predict the risk of MI and death [5,6]. The myocardial tissue-specific biomarkers of cardiac necrosis, including cardiac troponins (cTn) and heart-type fatty acid binding protein (H-FABP), help diagnose MI in the early hours following symptoms and distinguish between MI and UA [7,8,9,10]. Novel prognostic biomarkers of CAD and ACS reflect different aspects of atherosclerosis development: inflammation, fibrosis, plaque instability, platelet activation, neurohormonal activation, and myocardial stress [5]. A number of protein biomarkers, including natriuretic peptides, soluble suppression of tumorigenicity 2 (sST2), growth differentiation factor-15 (GDF-15), high-sensitivity cardiac troponins (hs-cTn), C-reactive protein, interleukin 6, and myeloperoxidase, as well as multi-marker approaches, have the potential to estimate the risk of major adverse cardiac events (MACE) in patients with stable CAD [11]. 

MicroRNAs (miRNAs) are a class of small RNAs that are extraordinarily stable in biofluids, as they are incorporated into extracellular microvesicles (MVs) or bound with circulating proteins and thus are protected from RNAse activity [12]. Circulating extracellular miRNAs are considered to be promising non-invasive biomarkers for many pathologies, including cardiovascular disease (CVD) [13,14,15,16,17]. A significant number of recent studies have considered circulating extracellular miRNAs as potential blood-based biomarkers for ACS diagnostics and prediction (reviewed in References [18,19,20]). The most frequently reported miRNAs in all types of CAD (miR-1, miR-133a, miR-208a/b, and miR-499) are expressed abundantly in the heart and play crucial roles in cardiac physiology [20]. Many miRNAs participate in the regulation of atherosclerosis development [21]. As for protein biomarkers of CAD and ACS, changes in the miRNA circulating levels might occur as a result of various processes associated with atherogenesis. 

In this study, we analyzed plasma levels of circulating miRNAs in patients with ACS (UA, STEMI, and NSTEMI) and stable CAD compared to healthy controls and patients with arterial hypertension (HT) without CAD. The analysis included miRNAs potentially involved in atherosclerosis development, stably detectable in plasma of healthy adults (among the top 200 abundant plasma miRNAs), and known to be up- or downregulated in the plasma of patients with CAD or ACS. Candidate miRNAs included two species from the 17/92a cluster (miR-17-5p and miR-92a-3p) and miR-126-3p, which are expressed in endothelial cells and regulate vascular integrity and angiogenesis [22,23,24,25]. These miRNAs were recently discovered as potential CAD circulating biomarkers [26,27,28]. Two inflammatory miRNAs were included in the study: miR-155-5p, linked with inflammation and immunity [29], and miR-146a-5p, known as a regulator of innate immune responses [30]. It has been shown that an elevated plasma level of miR-146a is a potential biomarker of AMI [31]. On the contrary, a lowered level of miR-155-5p was recently discovered in CAD patients, compared to controls [32]. The most abundant miRNA in normal arteries and vascular smooth muscle cells (VSMCs), miR-145-5p, is involved in VSMC phenotypic modulation and proliferation and controls vascular neointimal lesion formation [33,34]. This miRNA has recently been found to be downregulated in the plasma of CAD patients [32]. Often mentioned in connection with atherosclerosis, miR-21-5p is linked with VSMC proliferation and apoptosis, cardiac cell growth and death, as well as cardiac fibroblast functions [35]. Expression of miR-21-5p is deregulated in the heart and vasculature under cardiovascular disease conditions, and it appears to have the potential as a biomarker to differentiate the diagnosis of ACS from stable CAD [20]. Recently it was shown that the plasma level of miR-21-5p is elevated in patients with ACS compared to patients with stable angina and control subjects [36]. Additionally, miR-375-3p was included in the analysis because it is altered in the plasma of MI patients according to the results of circulating miRNA sequencing studies and regulates PIK3CA and TP53 genes—the key players in the MI disease module [37]. The main goal of this study was to analyze whether circulating plasma levels of these CVD-related miRNAs differ between CAD patients and non-CAD individuals, between ACS and stable CAD patients, and between troponin-negative (UA) and troponin-positive (MI) patients within the ACS group.

Pre-analytical and methodological differences in previous studies are probably the cause of the discrepancies existing for several miRNAs in terms of their potential as circulating biomarkers for CVD [38,39,40,41,42,43,44]. Furthermore, medication therapies affecting the platelet state may alter the circulating miRNA profile as a result of platelet activation and the release of platelet-derived MVs [45]. As the measurement of blood circulating extracellular miRNA content is influenced by a number of pre-analytical factors, we fulfilled the basic pre-analytical requirements provided for circulating extracellular miRNA research in this study. This was accomplished through the choice of plasma instead of serum as the blood-derived extracellular biofluid, a standardized two-step centrifugation process for platelet-free plasma isolation, accurate control of sample hemolysis, and the use of stable endogenous plasma miRNA as a normalization control for relative miRNA level measurements. The possible influence of different medication therapies received by patients on circulating miRNA levels was also estimated in this study.

## 2. Materials and Methods

### 2.1. Study Population

University cardiology clinic patients (males and females from 40 to 90 years old) were included in the study. Several patient groups were recruited: patients with ACS (UA, STEMI and NSTEMI) with the exception of those meeting the exclusion criteria; patients with stable CAD with the exception of those meeting the exclusion criteria; patients with hypertension (HT) without CAD from the outpatient department; and healthy controls. UA, STEMI, NSTEMI, stable CAD, and HT were diagnosed according to European Society of Cardiology criteria. The exclusion criteria were severe concomitant pathology (end-stage lung, kidney, or liver disease; diabetes decompensation), any active malignancy, alcohol and drug abuse, or inability or unwillingness to provide written consent.

### 2.2. Plasma Collection and Storage

Whole blood was collected in 6 mL K2-EDTA tubes (BD, Franklin Lakes, NJ, USA) and stored at room temperature for no more than one hour prior to plasma isolation. For MI patients, blood was taken within a period from several hours to 7 days after MI (mean value: 2.22 days). To obtain platelet-free plasma (PFP) for miRNA isolation, the whole blood sample was processed in two steps—whole blood was centrifuged at 2130× *g* for 10 min at room temperature (Step 1), and then the crude plasma was repeatedly centrifuged in sterile 15 mL conical tubes at 2130× *g* for 10 min at room temperature (Step 2). The upper two-thirds of the plasma layer was stored at −20 °C in 500 μL portions in sterile RNAse-free 1.5 mL tubes in the clinical facility. Within 1 month, frozen plasma samples were transported to the laboratory facility without thawing and then stored at −80 °C. Isolation of miRNA was performed within one month after blood collection. Before miRNA isolation, plasma samples were thawed on ice and centrifuged at 16,000× *g* for 15 min at 4 °C to pellet down any residual cell debris (Step 3). The supernatant in a volume of 300 µL was used immediately for miRNA extraction and 10 µL was aliquoted and stored at −20 °C for further hemolysis assessment.

### 2.3. Hemolysis Assessment of Plasma Samples

Only samples without visually detected hemolysis were included in this study. Low-level red blood cell (RBC) hemolysis in plasma samples was assessed via the spectrophotometric measurement of absorbance at the 414 nm wavelength (peak of free hemoglobin). For each sample, a 10 µL aliquot of plasma supernatant from centrifugation Step 3 was used for hemolysis assessments within seven days after miRNA isolation. The sample was thawed, incubated at room temperature for 30 min, and analyzed on NanoDrop^®^ 2000 spectrophotometer (Thermo Fisher Scientific, Waltham, MA, USA) via the measurement of ultraviolet–visible (UV–vis) absorbance with a 1 mm path at 385 nm (A385) and 414 nm (A414) wavelengths in triplicate for each sample. For each measurement, a lipemia-independent hemolysis score (HS) was calculated based on the mean A414 and A385 values: HS = ∆(A414 − A385) + 0.16 ∗ A385 [46]. Samples with HS > 0.25 were not included in this study. 

### 2.4. Plasma miRNA Isolation

First, miRNA was isolated from 300 µL of the plasma supernatant from centrifugation Step 3, using a NucleoSpin miRNA Plasma kit (Macherey-Nagel, Düren, Germany) according to the manufacturer’s guidelines. Proteinase K digestion was performed for each plasma supernatant sample before the isolation of miRNA. Each miRNA sample had a total volume of 30 µL and was stored at −80 °C prior to cDNA synthesis.

### 2.5. cDNA Synthesis and Quantitative PCR (qPCR) for miRNA Detection

A 2 µL sample of miRNA was used for cDNA synthesis with a TaqMan Advanced miRNA cDNA Synthesis Kit (Thermo Fisher Scientific, Waltham, MA, USA), according to the manufacturer’s recommendations, using a DNA Engine Tetrad 2 Thermal Cycler (Bio-Rad Laboratories, Hercules, CA, USA). Eleven commercially available TaqMan Advanced miRNA assays with TaqMan Fast Advanced Master Mix (Thermo Fisher Scientific, Waltham, MA, USA) were used to perform qPCR, according to the manufacturer’s protocol. The list of miRNA assays with catalogue numbers and mature miRNA sequences is given in Table 1. The list includes the eight miRNAs involved in cardiovascular pathology and atherogenesis by regulating vascular integrity, angiogenesis, and VSMC proliferation, or linked with inflammation and immunity [21,22,23,24,25,29,30,31,32,33,34,35,47]. All these miRNAs had previously reported altered plasma levels in CAD or ACS patients [17,18,19,20]. Each miRNA had the mean log_10_ value of miR-16-5p-normalized relative plasma level not less than −2 in at least one of the two most recent publications which used extracellular plasma small RNA sequencing in a cohort of healthy subjects (Max et al., 2018 [48] and Godoy et al., 2018 [49]). As an endogenous normalization control, hsa-miR-16-5p was chosen, since it is widely used for plasma miRNA studies. For qPCR-based hemolysis assessment, a pair of miRNAs (hsa-miR-23a-3p and hsa-miR-451a) that indicate the degree of hemolysis by their ratio were included in the study. All of the analyzed miRNAs were amongst the most commonly found circulating miRNAs in plasma [50]. For each miRNA assay, a no-template control (NTC) containing nuclease-free water instead of a miRNA sample was analyzed. We performed qPCR by using the QuantStudio 5 Real-Time PCR system (Thermo Fisher Scientific, Waltham, MA, USA) in MicroAmp 96-well PCR plates and optical adhesive film, in a “Fast” cycling mode with the following program: enzyme activation—20 s at 95 °C; 45 cycles, denature—1 s at 95 °C, anneal/extend—20 s at 60 °C. We obtained qPCR data by using QuantStudio Design and Analysis Software v1.4.1 (Thermo Fisher Scientific, Waltham, MA, USA). Cq values were calculated by using the automatic “Baseline” value and the experimentally set “Threshold” value of ∆Rn = 0.15 for all analyzed miRNA targets. Cq measurements were performed in a single technical replicate for each miRNA target within an individual sample. Normalization of qPCR data for each target miRNA was performed by using the Cq value of hsa-miR-16-5p. For each miRNA analyzed, its plasma relative expression level was calculated as 2^−ΔCq (target miRNA–hsa-miR-16-5p)^. If the Cq value of the analyzed miRNA was undetectable by 45 qPCR cycles, its Cq was considered to be 45. The impact of RBC hemolysis on circulating miRNAs was estimated based on the qPCR-detected ratio of hemolysis-dependent miRNA miR-451a abundant in RBC and miR-23a-3p, which is hemolysis-independent [50]. The difference between Cq values of these miRNAs was calculated by the formula: dCq_miR-23a-3p–miR-451a_ = Cq(hsa-miR-23a-3p)–Cq(hsa-miR-451a). For all the samples in this study, the same laboratory workflow and PCR data analysis protocol was used.

### 2.6. Data Analysis

The Mann–Whitney U test was performed for pairwise comparisons of miRNA relative plasma levels and sample characteristics between the study groups. Differences with *p*-values of <0.05 were considered statistically significant. Multiple linear regression (MLR) analysis *p*-values were obtained with the Bonferroni–Holm correction for multiple comparisons; both HS and dCq_(miR-23a-3p—miR-451a)_ values, as well as the presence type 2 diabetes, were used as confounding factors; differences with *p*-values of <0.1 were considered statistically significant. Data analysis and visualization were performed by using SPSS Statistics software (version 26.0, Armonk, IBM, NY, USA) and the *tidyverse* R package (version 1.3.0).

## 3. Results

### 3.1. Study Sample Characteristics

A total of 136 subjects were included in this study, according to the clinical inclusion/exclusion criteria and plasma/miRNA sample quality requirements. The study sample included four groups: ACS: 50 patients with ACS, including 26 patients with UA and 24 patients with MI (13 STEMI and 11 NSTEMI patients);SCAD: 26 patients with stable CAD;HT: 30 hypertensive patients without CAD;CONTR: 30 healthy controls.

The main characteristics of the study sample groups are shown in Table 2. Age gaps between HT patients and controls and between CAD patients and controls were ~10 years and ~15 years, respectively. There were no statistically significant differences in age between the HT, SCAD, and ACS groups (Mann–Whitney test). The ACS group had a pronounced sex bias, with 80% men included in the group. Age and sex biases in the study groups were consistent with the global data on the age- and sex-specific prevalence and incidence of CAD and ACS [2]. Both the SCAD and ACS groups included ~25% of patients with type 2 diabetes mellitus (DM). The majority (~60%) of patients with UA from the ACS group had an MI history with one previous MI case from several months ago to 20 years ago (mean time from the last MI case: 7.3 years, SD = 7.3; mean age of the first MI case: 57.3 years, SD = 10.6). The majority of MI patients had plasma troponin T (TnT) levels from 0.1 to 4.5 ng/mL, with a mean value of 1.6, SD = 1.4, and two STEMI patients had severely elevated TnT levels (8.3 and 22.0 ng/mL). Distribution plots of the main characteristics in the study groups are shown in Figure A1, Appendix A. Mann–Whitney *p*-values of significance for pairwise comparisons of the main characteristics between the study groups with the Bonferroni–Holm correction are given in Table A1, Appendix A. Plasma low-density lipoprotein (LDL) levels were lower in healthy controls compared to all other groups (*p* < 0.05). Plasma high-density lipoprotein (HDL) levels were lower in ACS patients compared to all other groups (*p* < 0.05). 

### 3.2. Circulating Extracellular Plasma miRNA Levels

All miRNAs were detectable in plasma by means of the qPCR method with the mean Cq values from 13.9 for the most abundant miR-451a to 29.6 for the lowly-expressed miR-155-5p. To evaluate the consistency of our data with the present plasma whole miRNA profile studies, we compared the relative plasma levels of the target miRNAs with those obtained from the two most recent publications which used plasma small RNA sequencing in cohorts of healthy adults (Max et al. 2018 [48] and Godoy et al. 2018 [49]). These studies contained the miRNA sequencing (miRNA-seq) data obtained from plasma samples of 13 and 12 adults, respectively. For comparison with our qPCR data, relative plasma levels from sequencing data were calculated as normalized read counts of each miRNA related to read counts of miR-16-5p. The results of this comparison are presented in Figure 1. The concordance of the results between two sequencing studies for the analyzed miRNAs was higher than that found between our results and any of these studies. Though the direction of change for most of the miRNAs between each other was similar, our study showed significantly lowered miR-16-5p-normalized relative plasma levels for the majority of miRNAs compared to miRNA-seq data.

To estimate the influence of plasma sample hemolysis on circulating miRNAs, we analyzed the association between the spectrophotometric-based hemolysis index (HS) and the qPCR-based ratio between hemolysis-dependent miRNA pair (dCq_(miR-23a-3p– miR-451a)_ value). The results of the hemolysis assessment of plasma samples from the study groups are shown in Table 2. The SCAD group had lowered HS values, and the HT group had elevated dCq_(miR-23a-3p–miR-451a)_ values (*p* < 0.05, Mann–Whitney test), so both hemolysis indices were included in the analysis as additional confounding factors. HS showed an association and positive correlation with the dCq_(miR-23a-3p–miR-451a)_ in the whole sample (adjusted linear R^2^ = 0.177, *p* = 1.98 × 10^−7^; Spearman’s Rho 0.472, *p* < 0.05; Figure A2, Appendix A).

### 3.3. Extracellular Plasma miRNA Markers of SCAD and ACS

Principal component analysis (PCA) of relative plasma levels of eight miRNAs potentially related to CAD showed the best discrimination (~70 to 80% of cases) between the CONTR and ACS groups (Figure 2). PCA results are presented in Table A2, Appendix A. The HT and SCAD groups shared the majority of cases in PCA plot and were not clearly distinguishable among themselves, as well as between the CONTR and ACS groups.

Furthermore, we compared relative plasma levels of the analyzed miRNAs between the study groups by using the Mann–Whitney test for pairwise comparisons. Figure 3 shows boxplots of the miRNA relative plasma level distribution in the study sample groups, with markings indicating statistically significant differences between groups. Table 3 contains the results of group pairwise comparisons including fold change values and the presence of *p*-values less than 0.05, determined by the Mann–Whitney test with and without the Bonferroni–Holm correction for multiple comparisons. 

Patients with hypertension had lowered relative miR-17-5p plasma levels, compared to healthy controls. No statistically significant differences in the relative plasma levels of any of the other analyzed miRNAs were found between the CONTR and HT groups (Table 3).

Among the analyzed miRNAs, miR-21-5p and miR-146a-5p showed similar distributions of relative plasma levels in the study groups (Figure 3). Relative plasma levels of these miRNAs were slightly elevated in stable CAD patients compared to healthy controls. Plasma miR-21-5p was upregulated in stable CAD patients compared to patients with hypertension. ACS patients had more than two-fold elevated plasma levels of miR-21-5p and miR-146a-5p compared to all other groups (Figure 3 and Table 3). 

For miR-155-5p and miR-126-3p, a slight increase in the relative plasma level (1.5–1.75-fold) was shown in the ACS group compared to the HT and CONTR groups.

The plasma level of miR-17-5p showed a completely different distribution, being significantly lower in both ACS and SCAD groups compared to healthy controls and patients with hypertension (Figure 3 and Table 3). Patients with stable CAD had lowered relative miR-17-5p plasma levels compared to healthy controls and patients with hypertension. 

The plasma level of miR-92a-3p was slightly increased in ACS patients compared to patients with stable CAD and patients with hypertension (Table 3). However, no statistically significant differences in the miR-92a-3p relative plasma level were found between ACS patients and healthy controls.

A statistically significant downregulation of plasma miR-375-3p was observed for the ACS group compared to healthy controls.

To estimate whether miRNA relative plasma levels distinguished the study sample groups, considering the possible influence of hemolysis and the presence of type 2 diabetes, we performed multiple linear regression (MLR) analysis with the hemolysis indices and the presence of type 2 diabetes as confounding factors (Table 3). The results of the MLR analysis showed that the relative plasma level of miR-21-5p was an independent marker of ACS. Downregulated miR-17-5p distinguished ACS and stable CAD patients from patients with hypertension and healthy controls. Upregulated miR-146a-5p was a marker of ACS patients compared to stable CAD patients and healthy controls.

Summarizing the results, we can conclude that upregulated plasma miR-146a-5p and miR-21-5p are possible biomarkers of ACS, and downregulated plasma miR-17-5p might be considered as a biomarker of CAD, without differences between ACS and stable CAD. For other analyzed miRNAs, changes in ACS patients compared to other groups were not pronounced. 

Furthermore, we compared relative plasma levels of miR-146a-5p, miR-21-5p, and miR-17-5p between patients with UA and MI (STEMI and NSTEMI). There were no statistically significant differences between the compared groups (Mann–Whitney test, *p* > 0.05; Figure 4), indicating that altered levels of these miRNAs are markers of both troponin-negative (UA) and troponin-positive (MI) ACS patients. Within the MI group, there were no statistically significant differences in relative plasma levels of miR-146a-5p, miR-21-5p, and miR-17-5p between the STEMI and NSTEMI groups. Relative plasma levels of miR-146a-5p, miR-21-5p, and miR-17-5 did not correlate with the troponin T level in MI patients.

### 3.4. Influence of Age and Sex on Plasma miRNA Levels

To estimate the age contribution to the obtained results, we performed two additional variants of MLR analysis for a comparison of miRNA relative plasma levels in the study sample groups. First, we included age as an additional confounding factor in the MLR analysis of the comparison of miRNA relative plasma levels in the study sample groups. The significant differences, with *p* < 0.05, persisted for the ACS vs. CONTR comparison for miR-17-5p, miR-21-5p, and miR-375-3p; for the ACS vs. HT comparison for miR-17-5p and miR-21-5p; and for the ACS vs. SCAD comparison for miR-146a-5p and miR-21-5p. We further performed MLR analysis of the relative target miRNA plasma levels to obtain *p*-values of significance for age prediction without any clinical parameters as confounding factors. This analysis showed that miR-146a-5p, miR-17-5p, miR-21-5p and miR-375-3p plasma levels have a significant (*p* < 0.05) relationship with age.

Among the circulating miRNAs deregulated in ACS, no statistically significant differences in relative plasma levels were found between males and females within the group of CAD patients (N = 74). However, miR-17-5p and miR-375-3p were downregulated in males within the combined group of healthy controls and hypertensive patients (N = 60, 32 males, 28 females; Mann–Whitney test, *p* < 0.05).

### 3.5. Influence of Medication Therapies on Plasma miRNA Levels

To estimate the relationship between miRNA relative plasma levels and any medication therapies received by patients, we performed MLR analysis for each miRNA and each type of potentially influencing therapy in a combined HT + SCAD + ACS group (N = 106). The results of this analysis showed no statistically significant relationships between miR-17-5p, miR-146a-5p, and miR-21-5p plasma levels and any type of therapy (Table A3, Appendix A). The plasma level of miR-92a-3p was positively influenced by the use of anticoagulants.

Within the combined group of CAD patients (SCAD + ACS, N = 74), we compared relative plasma levels of the analyzed miRNAs between groups receiving or not receiving specific medication therapies, using the Mann–Whitney test (Table 4). Among the potential ACS-related miRNAs, miR-21-5p was upregulated in patients receiving diuretics, and mir-17-5p was downregulated in patients receiving anticoagulants.

## 4. Discussion

In this study, we performed an analysis of plasma miRNAs in patients with stable CAD and ACS, with a detailed evaluation of the main pre-analytical parameters required for the correct measurement of circulating miRNA biomarkers. Using hemolysis indices and the presence of type 2 diabetes as confounding factors in the statistical analysis, we observed altered relative plasma levels in patients with ACS or stable CAD compared to healthy controls or hypertensive patients without CAD for the majority of the eight analyzed miRNAs potentially involved in atherosclerosis development.

Several challenges impede the accurate detection and use of plasma and serum miRNAs. The preliminary discussion of the obtained data should be carried out in accordance with several recently discovered important findings concerning circulating extracellular miRNA studies:The measurement of plasma miRNA levels is sensitive to pre-analytical factors, including the plasma preparation protocol, plasma sample hemolysis, the method of miRNA detection, and the normalization strategy [39,40,41,42,43,44];In vivo or in vitro platelet activation substantially affects plasma miRNA levels due to the release of platelet-related MVs (PMVs)—the major fraction of circulating MVs [45,51,52];Antiplatelet drugs, anticoagulants, and other medication therapies affecting the platelet state may alter plasma miRNA profiles in subjects receiving these therapies [45,53].

In this study, we used platelet-free plasma (PFP) obtained by means of two-step centrifugation according to the method of Duttagupta et al. [54] for miRNA isolation. A single additional centrifugation step minimizes the level of contaminating cellular RNA in the plasma sample, preserving the expression of circulating miRNA species [54]. Another important pre-analytical problem is bias due to the effect of RBC hemolysis on circulating miRNA levels. In our study, this problem was of key importance, since we used miR-16-5p as a reference endogenous control for miRNA plasma level normalization. Plasma miR-16 levels show small differences between individuals with different physiological conditions but are significantly affected by the presence of RBC hemolysis [42,44]. In the absence of hemolysis, the level of miR-16-5p is sufficiently constant to serve as a normalizer [42]. A number of CAD-associated microRNAs analyzed in this study (miR-126-3p, miR-145-5p, miR-21-5p, and miR-92a-3p) are also sensitive to RBC hemolysis [42,44]. Therefore, accurate hemolysis control is a way to reduce the possible bias in miRNA detection. The spectrophotometric and qPCR-based hemolysis assessment was performed for all plasma samples included in this study. We found that the spectrophotometric-based HS index correlates with the hemolysis-dependent miR-23a-3p/miR-451a miRNA ratio, and established the following criteria for plasma sample inclusion: HS < 0.25 and dCq (miR-23a-3p–miR-451a) < 14. 

We used relatively novel TaqMan Advanced Assays with universal reverse transcription for qPCR-based miRNA detection, which is more convenient in the simultaneous analysis of dozens of miRNA targets than single TaqMan assays. We showed the presence of a substantial bias in relative miRNA plasma levels measured by qPCR in our sample compared to those obtained from the recent plasma miRNA-seq data in healthy adult samples for the majority of the analyzed miRNAs. This bias may be taken into account in any further qPCR validation of miRNA-seq results, especially considering that several research groups have recently started to use TaqMan Advanced miRNA assays in circulating miRNA studies of cardiovascular diseases [36,55].

In vivo or in vitro platelet activation could not be directly assessed within the design of this study, but we can suggest whether increased platelet activation occurred in CAD patient groups by estimating the relative plasma levels of platelet miRNAs, e.g., miR-126-3p—one the most abundant miRNAs in platelets [52]. The plasma level of miR-126-3p was only slightly increased in ACS patients compared to other study groups, which could be caused by both platelet activation and release from endothelial, myeloid, or lymphoid cells. Different distributions of relative plasma levels of other miRNAs contained in platelets (miR-17-5p, miR-21-5p, and miR-92a-3p) in the study groups (Figure 3) indicate that platelet activation was probably not the major reason for changes in miRNA plasma levels in the ACS and SCAD groups.

Previously, d’Alessandra et al. [56] discovered the a significantly (more than 10-fold) increased plasma relative miR-16-5p-normalized level of miR-126-3p in both stable and unstable CAD patients compared to healthy controls. However, no increase in the plasma level of miR-223-3p—the most abundant platelet-derived miRNA—was reported in CAD patients in that study, indicating that the elevated miR-126-3p was possibly not related to platelets. The majority of platelet miRNAs previously implicated with CVDs are not platelet-specific but present in various cells of the cardiovascular system [45]. At the moment, little is known about platelet MVs’ miRNA content and the possibility for the targeted release of miRNA-containing vesicles from platelets. Thus, it is worth considering that non-simultaneous elevations of platelet-abundant miRNAs such as miR-223-3p or miR-126-3p in circulation may be associated not only with platelet activation but may also be the result of impaired vascular function.

The main finding of our study is the significant increase (more than two-fold) in the relative plasma levels of circulating miR-146a-5p and miR-21-5p in patients with ACS compared to healthy controls, hypertensive patients without CAD, and patients with stable CAD. Furthermore, relative plasma levels of these miRNAs were higher in stable CAD patients compared to healthy controls and patients with hypertension, indicating that elevated miR-146a-5p and miR-21-5p may reflect general pathophysiological processes of CAD development. Both miR-146a-5p and miR-21-5p are among the most frequently reported miRNAs to be upregulated in ACS compared with stable CAD [20]. In our study, significantly elevated plasma levels of these miRNAs were general markers of any ACS, as they did not have statistically significant differences between patients with UA, STEMI, and NSTEMI. In the recent studies using TaqMan Advanced miRNA assays for circulating miRNA detection, the relative plasma level of miR-21-5p was elevated in both CAD and CHF patient groups compared to healthy controls [36,55]. Interestingly, elevated circulating levels of both miR-21-5p and miR-146a-5p were previously mentioned in disease states linked with inflammation [57,58]. As an inflammatory process is implicated in the pathogenesis of ACS, inflammation may be the reason for the upregulation of these miRNAs in the plasma of ACS patients. However, other inflammation-related miRNAs were not increased (miR-145-5p) or were only slightly increased (miR-155-5p) in the plasma of ACS patients in our study.

Unlike the miRNAs mentioned above, miR-17-5p plasma levels were significantly lower in ACS and stable CAD patients compared to healthy controls (more than two-fold) and hypertensive patients (~1.6-fold). Lowered plasma miR-17-5p in our sample was the general marker of CAD due to the lack of differences in plasma miR-17-5p between the ACS and SCAD groups. The existing data on miR-17-5p as a biomarker of CAD and ACS is controversial. Fichtlscherer et al. [59] showed that the circulating levels of miR-17-5p, as well as those of miR-126-3p, miR-92a-3p, and miR-155-5p were significantly reduced in patients with CAD compared with healthy controls. Xue et al. [27] reported significantly increased plasma levels of miR-17-5p in MI patients compared to healthy controls. Both these studies used a similar protocol for plasma preparation and cel-miR-39 as a normalization control for qPCR-based miRNA quantification. Another two studies, which used U6 small nuclear RNA as a normalization reference for circulating miRNA qPCR measurement, also provide controversial data on miR-17-5p: Zhang et al. [60] reported remarkably decreased plasma levels of miR-17-5p in patients with CHD, whereas Chen et al. [61] showed that an elevated plasma level of miR-17-5p is a marker of CAD and reflects the severity of atherosclerosis in CAD patients. However, it should be noted that the results of U6-normalized circulating miRNA studies are questionable as U6 is not suitable as a reference for total plasma/serum miRNA studies [62].

Elevated circulating miR-92a-3p in our study distinguished ACS patients from patients with stable CAD and hypertension. However, the lack of statistically significant differences in the miR-92a-3p plasma level between ACS patients and healthy controls does not allow us to consider it an ACS marker in our study. Recently it has been shown that miR-92a-3p was significantly increased in circulating plasma MVs of stable CAD and ACS patients compared with non-CAD individuals [26]. The analysis of miRNA content in different MV subpopulations separated by flow cytometry revealed that endothelial cell-derived MVs are enriched in miR-92a-3p. In our study, the whole circulating extracellular miRNA pool was analyzed, including exosome- and MV-derived miRNAs and miRNA-protein complexes. Patients with CAD show increased plasma levels of circulating MVs as a consequence of endothelial activation and dysfunction [63], and we can only speculate as to whether elevated plasma miR-92a-3p in ACS patients in our study was the result of the increased endothelial-derived MV release. 

Circulating plasma miR-375-3p was decreased in ACS patients compared to healthy controls in our study. This result is consistent with previous qPCR-based studies [56] and a recent extracellular miRNA-seq-based study by Baulina et al. [37] showing downregulated miR-375 in plasma of MI patients the day after disease onset. 

The possible effect of medication therapies on circulating miRNA levels was estimated in this study. As the majority of the CAD patients (above 75%) received antiplatelet drugs and statins, these types of therapies may be the collateral cause of a change in the circulating plasma levels of some miRNAs in CAD patients compared to other study groups. Within the group of CAD patients, some miRNAs associated with ACS in our sample had the same direction of change in patients receiving ACEI (miR-375-3p), diuretics (miR-21-5p), and anticoagulants (miR-17-5p). However, CAD patients not receiving diuretics still showed elevated miR-21-5p plasma levels, and patients not receiving anticoagulants showed lowered miR-17-5p plasma levels compared to controls. 

A specific source of circulating microRNAs cannot be explicitly established in studies that measure the total pool of circulating microRNAs without dividing into fractions (exosome- or MV-derived, HDL-associated, or bound with circulating proteins). However, we can speculate about the most likely potential source for several circulating miRNAs, as some are tissue-specific or abundant in specific fractions found in circulation. The group of miRNAs which are most abundant in the skeletal or heart muscle (so-called myomiRs) include miR-1-3p, miR-133a-3p, miR-133b, miR-208a/b-3p, and miR-499a-5p. These miRNAs have very low plasma/serum extracellular levels in healthy individuals, but these levels increase after intense physical exercise, as well as after myocardial infarction [20,64,65,66]. Similar to troponins, circulating levels of myomiRs may reflect the severity of myocardial damage under ischemic conditions. The circulating level of liver-specific miR-122-5p is increased in patients with liver injury and can be used to assess impaired liver function in patients with CVD [67]. Platelet-abundant miRNAs, such as miR-223-3p and miR-126-3p, may reflect changes in the platelet state caused by CVD or by several medication therapies, but their origin in circulation may not only be platelet-derived, as mentioned above. Some of CAD-associated miRNAs analyzed in this study are abundant in plasma-derived CD63+/CD9+ exosomes—miR-126-3p, miR-92a-3p, miR-21-5p, and miR-146a-5p [68]. Among them, miR-126-3p and miR-92a-3p, as well as miR-145-5p, have been found in MVs released by endothelial cells [26,69,70]. MiR-145-5p are normally abundant in VSMC but can be rapidly depleted in atherosclerotic conditions [71,72]. The uptake of EC-derived MVs containing miR-145-5p, miR-126-3p or miR-92a-3p by the recipient smooth muscle or endothelial cells provides atheroprotective effects in vivo [26,69,73]. Thus, changes in the circulating extracellular levels of these miRNAs may be caused by endothelial secretion of MVs due to impaired vascular function.

This study shared control groups (the CONTR and HT group) with our previous study on potential circulating miRNA markers of paroxysmal atrial fibrillation (PAF) [74]. Although the list of the analyzed miRNAs differed between the two studies, the main principles of the study design, as well as the methods of plasma collection and qPCR miRNA detection, were the same. Therefore, the limitations of this study were similar to those of the PAF study, which were described previously [74]. 

Considerable age and sex biases between CAD patients and healthy controls existed in this study and were expected as the result of the initial study design with the described inclusion and exclusion criteria for the sample collection and the formation of the study groups. The present study cannot be considered properly case–control age- and sex-matched. The fact that aging is the dominant risk factor for clinically significant atherosclerotic lesion formation explains the relationship with age for circulating miRNAs associated with CAD (miR-146a-5p, miR-17-5p, miR-21-5p, and miR-375-3p). 

Another important limitation of this study was the difference in time intervals between the onset of MI and blood collection for MI patients (from several hours to several days). Plasma levels of some miRNAs, especially cardiac-specific ones, may change dramatically during the post-infarction period [75]. However, this limitation could not have been overcome in the existing clinical conditions due to ethical issues of patient care.

The number of miRNA markers analyzed by using qPCR in this study was limited by study funding. Liver-specific miR-122-5p, of which the plasma level may reflect impaired liver function, and platelet-enriched miR-223-3p, indicating the level of platelet activation, were not included in this study. These miRNAs have been recently discovered to be circulating biomarkers for plaque instability and their relative serum levels were elevated in patients with CAD [76]. Undoubtedly, the inclusion of some additional microRNAs linked with atherosclerosis in the analysis would have increased the informativeness and value of this study.

As an improvement on our previous PAF study design, in this study miR-126-3p—one of the most abundant miRNAs in platelets—was included in the analysis, which made it possible to estimate whether platelet activation influenced the miRNA profile in CAD patients.

In conclusion, despite several limitations, this study was carried out in accordance with the main current recommendations for circulating miRNA research and reveals the utility of previously known CAD-associated circulating miRNAs as potential biomarkers for ACS and stable CAD. Using data on the influence of pre-analytical and analytical conditions, the presence of concomitant diseases, the use of medication therapies, and the biological role and origin of extracellular circulating miRNAs, we investigated and discussed the potential background for the observed miRNA plasma level changes in CAD patients.

## Figures and Tables

**Figure 1 biomolecules-11-00962-f001:**
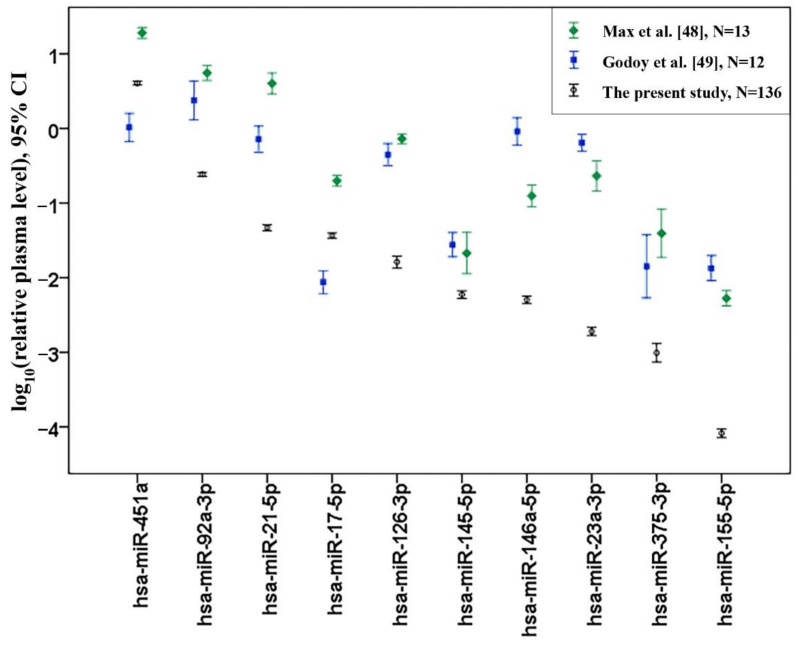
Comparison of miR-16-5p normalized relative plasma levels between the present study and the two most recent miRNA plasma sequencing studies performed on healthy adults. Error bars represent mean and 95% confidence interval (CI).

**Figure 2 biomolecules-11-00962-f002:**
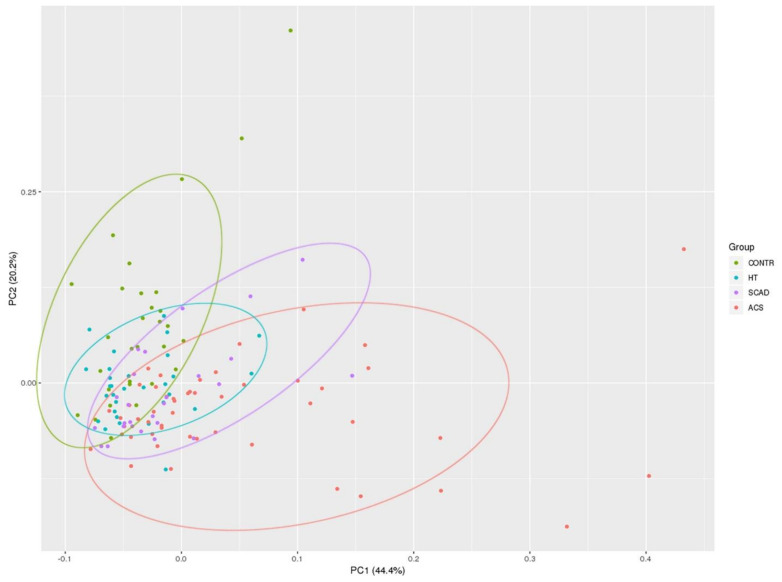
Principle component analysis (PCA) plot of relative plasma levels of eight analyzed miRNAs potentially associated with CAD.

**Figure 3 biomolecules-11-00962-f003:**
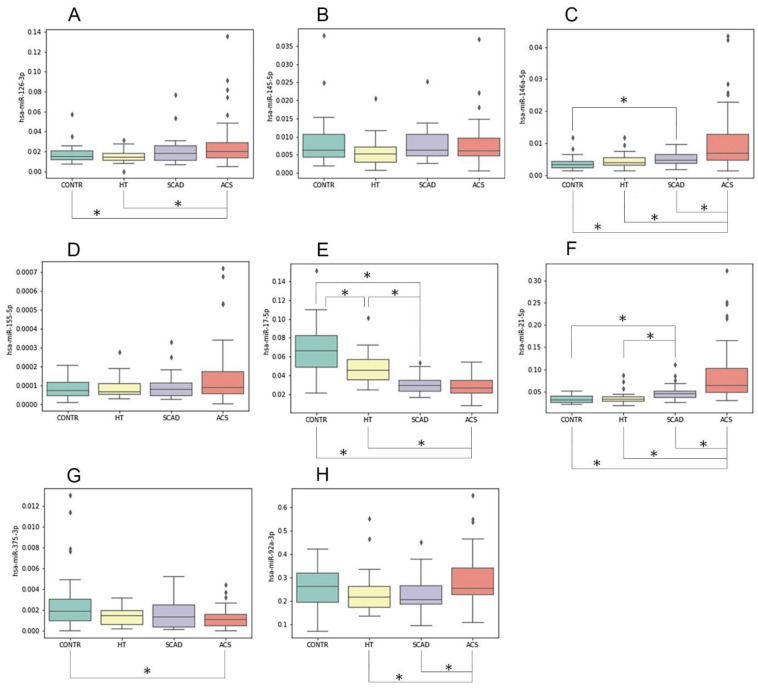
Relative miRNA plasma levels in the study sample groups: ACS, 50 ACS patients; SCAD, 26 patients with stable CAD; HT, 30 hypertensive patients without CAD; CONTR, 30 healthy controls. On the *y*-axis, relative plasma levels (normalized to miR-16-5p) are shown for miRNAs: miR-126-3p (**A**); miR-145-5p (**B**); miR-146a-5p (**C**); miR-155-5p (**D**); miR-17-5p (**E**); miR-21-5p (**F**); miR-375-3p (**G**); miR-92a-3p (**H**). The boxplots represent median and interquartile ranges (IQRs) in the box, minimum and maximum values in the “whiskers”, and outliers in the dots. Asterisks indicate differences in relative miRNA plasma level in the group pairwise comparisons (Mann–Whitney test, * *p* < 0.05).

**Figure 4 biomolecules-11-00962-f004:**
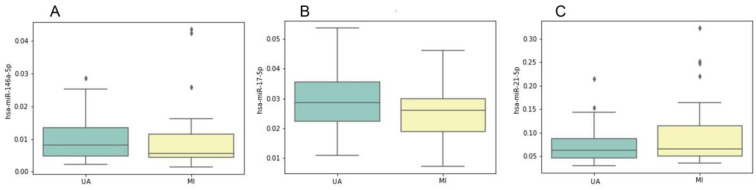
Relative miRNA plasma levels in the study sample groups within the ACS group (n = 50) for miR-146a-5p (**A**), miR-17-5p (**B**) and miR-21-5p (**C**). UA, 26 patients with unstable angina; MI, 24 patients with myocardial infarction (STEMI + NSTEMI). On the *y*-axis, relative miRNA plasma levels (normalized to miR-16-5p) are shown. The boxplots represent median and interquartile ranges (IQRs) in the box, minimum and maximum values in the “whiskers”, and outliers in the dots.

**Table 1 biomolecules-11-00962-t001:** List of the miRNA assays used for qPCR.

Assay Name	Assay ID	Mature miRNA Sequence	Type of miRNA
hsa-miR-16-5p	477860_mir	UAGCAGCACGUAAAUAUUGGCG	Normalization control
hsa-miR-23a-3p	478532_mir	AUCACAUUGCCAGGGAUUUCC	Hemolysis assessment
hsa-miR-451a	478107_mir	AAACCGUUACCAUUACUGAGUU	Hemolysis assessment
hsa-miR-126-3p	477887_mir	UCGUACCGUGAGUAAUAAUGCG	Candidate to ACS
hsa-miR-145-5p	477916_mir	GUCCAGUUUUCCCAGGAAUCCCU	Candidate to ACS
hsa-miR-146a-5p	478399_mir	UGAGAACUGAAUUCCAUGGGUU	Candidate to ACS
hsa-miR-155-5p	477927_mir	UUAAUGCUAAUCGUGAUAGGGGUU	Candidate to ACS
hsa-miR-17-5p	478447_mir	CAAAGUGCUUACAGUGCAGGUAG	Candidate to ACS
hsa-miR-21-5p	477975_mir	UAGCUUAUCAGACUGAUGUUGA	Candidate to ACS
hsa-miR-375-3p	478074_mir	UUUGUUCGUUCGGCUCGCGUGA	Candidate to ACS
hsa-miR-92a-3p	477827_mir	UAUUGCACUUGUCCCGGCCUGU	Candidate to ACS

**Table 2 biomolecules-11-00962-t002:** Characteristics of the study sample groups. CONTR, 30 healthy controls; HT, 30 hypertensive patients without CAD; SCAD, 26 patients with stable CAD; ACS, 50 patients with acute coronary syndrome (26 with unstable CAD and 24 with MI); CAD, coronary artery disease; MI, myocardial infarction; SD, standard deviation; HT, hypertension; DM, diabetes mellitus; AF, atrial fibrillation; ACEI, angiotensin-converting enzyme inhibitors; LDL, low-density lipoproteins; HDL, high-density lipoproteins; HS, hemolysis score.

Group ID	CONTR	HT	SCAD	ACS	Total
Number of patients	30	30	26	50	136
Mean age (SD), years	47.3 (5.6)	57.7 (9.5)	64.9 (7.5)	61.1 (10.9)	58.1 (10.8)
Gender (male/female)	15/15	17/13	15/11	40/10	87/49
Type 2 DM presence	0	0	6	13	19
Hyperlipidemia	0	7	22	35	64
HT presence	0	30	24	42	96
AF presence	0	0	1	2	3
MI history	0	0	0	16	16
Coronary atherosclerosis	0	0	15	34	49
**Blood lipid Profile**					
Total cholesterol, mmol/L, mean (SD)	4.55 (0.9)	5.19 (0.98)	5.34 (1.25)	4.95 (1.37)	4.96 (1.18)
Triglycerides, mmol/L, mean (SD)	1.13 (0.31)	1.46 (0.5)	1.61 (0.55)	1.81 (0.72)	1.53 (0.62)
LDL cholesterol, mmol/L, mean (SD)	2.14 (0.56)	2.69 (0.67)	2.98 (1.03)	2.98 (1.16)	2.7 (0.96)
HDL cholesterol, mmol/L, mean (SD)	1.86 (0.51)	1.84 (0.5)	1.53 (0.59)	1.06 (0.28)	1.51 (0.57)
**Medication Therapies**					
Beta-blockers	2	15	23	24	64
Calcium channel blockers	0	4	13	4	21
ACEI	0	12	12	32	56
Diuretics	0	4	9	16	29
Nitrates	0	1	9	2	12
Antiplatelet drugs	0	9	17	42	68
Anticoagulants	0	3	9	16	28
Statins	1	11	20	41	73
**Hemolysis Assessment of Plasma Samples**					
HS (SD)	0.122 (0.045)	0.144 (0.053)	0.103 (0.033)	0.121 (0.054)	0.123 (0.050)
dCq_(miR-23a-3p–miR-451a)_ (SD)	11.1 (0.98)	11.58 (1.09)	11.03 (1.12)	10.72 (1.22)	11.05 (1.16)

**Table 3 biomolecules-11-00962-t003:** Results of the study sample group pairwise comparisons to determine statistically significant differences in relative plasma miRNA levels. Different types of statistical analyses were used to obtain *p*-values of significance: Mann–Whitney test (M–W); Mann–Whitney test with the Bonferroni–Holm correction for multiple comparisons (M–W_corr); multiple linear regression analysis (MLR) with the Bonferroni–Holm correction for multiple comparisons, in which relative plasma levels of eight candidate miRNAs were included in the MRL model with HS, dCq(miR-23a-3p–miR-451a) and type 2 diabetes presence as confounding factors. Asterisks * indicate the presence of statistically significant changes between the compared groups. FC, fold-change values of miRNA mean relative plasma levels for pairwise comparisons; ACS, 50 ACS patients; SCAD, 26 patients with stable CAD; HT, 30 hypertensive patients without CAD; CONTR, 30 healthy controls.

miRNA	Type of Comparison	Group Comparison					
		ACS vs. CONTR	ACS vs. HT	ACS vs. SCAD	SCAD vs. CONTR	SCAD vs. HT	HT vs. CONTR
miR-126-3p	FC	1.53	1.75	1.26	1.21	1.39	0.87
	*p* < 0.05 (M–W)		*				
	*p* < 0.05 (M–W_corr)						
	*p* < 0.1 (MLR)						
miR-145-5p	FC	0.93	1.36	0.99	0.94	1.37	0.68
	*p* < 0.05 (M–W)						
	*p* < 0.05 (M–W_corr)						
	*p* < 0.1 (MLR)						
miR-146a-5p	FC	2.7	2.25	1.97	1.37	1.14	1.2
	*p* < 0.05 (M–W)	*	*	*	*		
	*p* < 0.05 (M–W_corr)	*	*				
	*p* < 0.1 (MLR)	*		*	*		
miR-155-5p	FC	1.8	1.72	1.61	1.12	1.07	1.05
	*p* < 0.05 (M–W)						
	*p* < 0.05 (M–W_corr)						
	*p* < 0.1 (MLR)						
miR-17-5p	FC	0.42	0.6	0.93	0.45	0.64	0,7
	*p* < 0.05 (M–W)	*	*		*	*	*
	*p* < 0.05 (M–W_corr)	*	*		*	*	
	*p* < 0.1 (MLR)	*	*		*	*	*
miR-21-5p	FC	2.69	2.42	1.79	1.5	1.36	1.11
	*p* < 0.05 (M–W)	*	*	*	*	*	
	*p* < 0.05 (M–W_corr)	*	*	*	*	*	
	*p* < 0.1 (MLR)	*	*	*		*	
miR-375-3p	FC	0.45	0.94	0.79	0.57	1.18	0.48
	*p* < 0.05 (M–W)	*					
	*p* < 0.05 (M–W_corr)	*					
	*p* < 0.1 (MLR)	*				*	
miR-92a-3p	FC	1.15	1.25	1.27	0.91	0.98	0.93
	*p* < 0.05 (M–W)		*	*			
	*p* < 0.05 (M–W_corr)		*	*			
	*p* < 0.1 (MLR)						

**Table 4 biomolecules-11-00962-t004:** Results of relative plasma miRNA level comparisons between groups receiving or not receiving specific medication therapies, using the Mann–Whitney test, within the combined SCAD + ACS group (n = 74). ↓ and ↑ symbols indicate lowered and elevated relative miRNA plasma levels in patients receiving therapy compared to patients not receiving therapy, respectively.

Medication Therapy	Number of Patients		Statistically Significant Change in Relative Plasma miRNA Level (*p* < 0.05)
	Not Receiving Therapy	Receiving Therapy	
Beta-blockers	29	47	miR-92a-3p↓
Calcium channel blockers	59	17	−
ACEI	32	44	miR-375-3p↓
Diuretics	51	25	miR-126-3p↑, miR-21-5p↑
Nitrates	65	11	−
Antiplatelet drugs	17	59	−
Anticoagulants	51	25	miR-17-5p↓
Statins	15	61	miR-126-3p↓

## Data Availability

The data are available from the corresponding author upon request.

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
