# Peer review of "Circulating Extracellular miRNA Analysis in Patients with Stable CAD and Acute Coronary Syndromes"

_biomolecules, 2021, doi:10.3390/biom11070962_

Round 1
Reviewer 1 Report
This article shows some methodological and design flaws.
miRNAs selection is not well justified and is based on several publications mostly older than 10 years old, although great advances in the miRNAs area have been done. More importantly, the patient's design is not based on the current classification of the ESC (references non included). STEMI and non STEMI (includes UA) and most of the patients of these group are hypertensive patients.
Reviewer 2 Report
Authors selected very carefully the miRNAs to evaluate by qPCR and potentially justify their usage as biomarkers. However, the explanation of the results is a bit complicated, and would recommend to try to reduce and simplify the sections. Also I would recommends authors explain better and divide in columns the different comparisons made in table 3.
The authors very nicely take into account the hemolysis in the samples, plus the selection of the reference miRNA seems very well founded.
I would also suggest to do a final working model of miRNAs secreted by the platelets / heart / endothelial cells that authors propose as biomarkers to better depict which miRNAs work for every type of CVD.
Author Response
Please see the attachment.

This manuscript is a resubmission of an earlier submission. The following is a list of the peer review reports and author responses from that submission.